# Quantitative MR Neurography in Multifocal Motor Neuropathy and Amyotrophic Lateral Sclerosis

**DOI:** 10.3390/diagnostics13071237

**Published:** 2023-03-25

**Authors:** Olivia Foesleitner, Karl Christian Knop, Matthias Lindenau, Fabian Preisner, Philipp Bäumer, Sabine Heiland, Martin Bendszus, Moritz Kronlage

**Affiliations:** 1Department of Neuroradiology, Heidelberg University Hospital, Im Neuenheimer Feld 400, 69120 Heidelberg, Germany; 2Neurologie Neuer Wall Dr. Bredow & Partner, Neuer Wall 19, 20354 Hamburg, Germany; 3dia.log, Altoetting Center for Radiology, 84503 Altoetting, Germany

**Keywords:** diagnostic imaging, diffusion tensor imaging, diffusion magnetic resonance imaging, peripheral nervous system diseases, amyotrophic lateral sclerosis

## Abstract

Background: The aim of this study was to assess the phenotype of multifocal motor neuropathy (MMN) and amyotrophic lateral sclerosis (ALS) in quantitative MR neurography. Methods: In this prospective study, 22 patients with ALS, 8 patients with MMN, and 10 healthy volunteers were examined with 3T MR neurography, using a high-resolution fat-saturated T2-weighted sequence, diffusion-tensor imaging (DTI), and a multi-echo T2-relaxometry sequence. The quantitative biomarkers fractional anisotropy (FA), radial and axial diffusivity (RD, AD), mean diffusivity (MD), cross-sectional area (CSA), T2-relaxation time, and proton spin density (PSD) were measured in the tibial nerve at the thigh and calf, and in the median, radial, and ulnar nerves at the mid-upper arm. Results: MMN showed a characteristic imaging pattern of decreased FA (*p* = 0.018), increased RD (*p* = 0.014), increased CSA (*p* < 0.001), increased T2-relaxation time (*p* < 0.001), and increased PSD (*p* = 0.025) in the upper arm nerves compared to ALS and controls. ALS patients did not differ from controls in any imaging marker, nor were there any group differences in the tibial nerve (*p* > 0.05). Conclusions: MMN shows a characteristic pattern of quantitative DTI and T2-relaxometry parameters in the upper-arm nerves, primarily indicating demyelination. Peripheral nerve changes in ALS seem to be below the detection level of current state-of-the-art quantitative MR neurography.

## 1. Introduction

Multifocal motor neuropathy (MMN) is a peripheral nerve disorder causing steadily progressive motor weakness in middle-aged patients, leading to high individual disability. The pathophysiology is not yet fully understood and presumably involves an immune-mediated dysfunction at the nodes of Ranvier. Treatment with intravenous immunoglobulins (IVIGs) is effective in 70–90%, but only if given in a timely manner, as untreated disease duration is the strongest predictor of worse outcome [1,2,3,4]. Definite diagnosis, however, may be difficult due to overlapping clinical symptoms, especially with amyotrophic lateral sclerosis (ALS), which implies a fatal diagnosis and completely different treatment strategy. Furthermore, most patients with MMN need IVIG maintenance therapy to prevent or slow down further motor weakness, and this necessitates a tailored treatment plan [5].

Previous imaging studies have shown good diagnostic accuracy for the differentiation of MMN and ALS based on the increased cross-sectional area (CSA) in the nerve ultrasound or nerve swelling, T2-hyperintensity, and comparison with the muscular denervation pattern in MR neurography [6,7,8]. However, CSA and T2-signal behavior provide only gross structural information and cannot assess microstructural changes occurring during disease progression [4,9,10].

Advanced MR methods could help to overcome this limitation. Diffusion-tensor imaging (DTI) and T2-relaxometry allow us to quantitatively assess nerve fiber integrity, as shown in various inflammatory and genetic/neurodegenerative neuropathies [11,12,13,14,15,16]. Moreover, DTI-derived parameters may potentially distinguish between axonal and myelin damage and detect remyelination [17]. Hence, quantitative MR sequences may provide non-invasive biomarkers for treatment monitoring and a better understanding of the pathophysiology throughout the course of the disease. So far, quantitative imaging studies in MMN and ALS are scarce and showed divergent results [18,19,20,21]. In particular, the characteristic phenotype of MMN and ALS, as depicted by quantitative MR neurography biomarkers, is still unclear.

The aim of this study was to characterize peripheral nerve changes and, thus, the underlying pathophysiology, as reflected by quantitative MR neurography (DTI, T2-relaxometry, and conventional nerve CSA) in patients with MMN and ALS compared to each other and to a group of healthy controls.

## 2. Materials and Methods

### 2.1. Participants

This study was approved by the institutional ethics committee and conducted in accordance with the Declaration of Helsinki. Written informed consent was obtained from all participants. Patients with ALS or MMN were part of a previous investigation [6] and prospectively recruited from one neurologic center and were examined by a neurologist specialized in peripheral nerve disorders with more than 25 years of experience. Inclusion criteria were an age between 20 and 80 years, a diagnosis of ALS (clinically definite or probable according to the Awaji criteria [22]), or of MMN (clinically definite or probable according to the criteria formulated by the European Federation of Neurological Societies and Peripheral Nerve Society criteria [23]). The clinical diagnosis was made while blinded to the MR neurography results. Additionally, healthy volunteers were recruited by public announcement. Exclusion criteria were any known neurologic or systemic disease except for ALS or MMN in patients and general contraindications for MRI. Demographic information was recorded for every participant, as well as detailed clinical/neurological characteristics and electrophysiological findings of patients, as described before and summarized in Appendix B Table A1 [6].

### 2.2. MR Neurography

All participants were examined on the same 3.0 Tesla MR scanner (Magnetom Skyra; Siemens Healthineers, Erlangen, Germany), using a 15-channel transmit/receive array coil (QED, Mayfield Village, OH, USA). The imaging protocol covered the mid-upper arm, the mid-thigh, and the proximal calf each, with three axial sequences acquired: (i) a high-resolution fat-saturated T2-weighted sequence for anatomical nerve delineation, (ii) a multi-spin-echo fat-saturated T2-relaxometry sequence, and (iii) a multi-shot echo planar imaging DTI sequence. Detailed sequence parameters are presented in Table 1. The lower extremity was examined in the supine position, while for the upper extremity, a prone position was taken, with the arm extended above the head. Patients were examined on the clinically more affected side. Five ALS patients did not tolerate positioning for the upper-arm examination.

### 2.3. Image Postprocessing

Image analysis was conducted in OsiriX MD version 12.5 (Pixmeo Sarl, Bernex, Switzerland). Due to variable intraneural connective tissue, only the tibial portion of the sciatic nerve was included, similar to previous MR neurography studies [24,25,26]. The tibial, median, radial, and ulnar nerves were manually segmented on six central, representative slices of the T2-weighted sequence to ensure comparability across subjects, and the cross-sectional area was measured. Regions of interest (ROI) were then transferred to the corresponding T2-relaxometry images and corrected for distortion and motion on the image with the highest echo time (120 ms). Mean signal intensities of the even echoes were then extracted, and an exponential function was fitted to receive the parameters of apparent T2-relaxation time (T2_app_) and protons spin density (PSD), as described previously [25,27]:$$S\left(TE\right)=PSD*e^{-\frac{TE}{{T2}_{app}}}+offset$$

Similarly, ROIs were transferred from the T2-weighted sequences to the b_0_-images of DTI and corrected for distortion. Maps of the scalar DTI parameters of fractional anisotropy (FA), mean diffusivity (MD), radial diffusivity (RD), and axial diffusivity (AD) were obtained using the plugin “DTI map” in OsiriX, with a fixed noise threshold of 10 arbitrary units (A.U.), referring to the signal intensity in the b_0_ image. This threshold was defined based on visual inspection of DTI-parameter maps with different noise thresholds (range 0–20). The mean nerve signal that was measured in the b_0_ image in six randomly selected participants was 74.2 A.U. (SD 14.8 A.U.), clearly exceeding the noise threshold. Subsequently, DTI parameters were extracted from the maps for all participants, and the mean values of six representative slices per slab were used for further analyses. In all, 6 out of the overall acquired 115 slabs were excluded after visual inspection due to relevant artifacts, of which 5 were at the calf (1 healthy subject, 2 ALS patients, and 2 MMN patients), and 1 was at the upper arm (healthy subject).

Due to the demographic differences between healthy participants and patients, we used multiple linear regression models to adjust the imaging parameters of the control cohort to 60 years of age and to a body weight of 70 kg to match patients based on previous publications [24,25]. Formulae are given in Appendix A.

### 2.4. Statistical Analysis

A statistical analysis was performed in SPSS version 28 (IBM, Armonk, NY, USA) and Prism version 9 (GraphPad Software, La Jolla, CA, USA). One-way analysis of variance with Holm–Sidak’s correction was used to assess group differences between patients with ALS or MMN and the healthy controls. For the statistical analysis, the values of the median, radial, and ulnar nerves were averaged. The significance level was set at *p* ≤ 0.05. The results are given as mean values ± standard deviation (SD) unless indicated otherwise.

## 3. Results

In total, 22 patients with ALS (male/female 12/10), 8 patients with MMN (male/female 7/1), and 10 healthy subjects (male/female 0/10) were included. The mean age was 62.3 years (SD 9.0) in ALS, 57.6 years (SD 18.6) in MMN, and 46.2 years (SD 13.4) in healthy volunteers (*p* = 0.007). The mean body weight was 69.5 kg (SD 11.9) in the ALS group, 78.9 kg (SD 11.2) in the MMN group, and 60.6 kg (SD 7.9) in the control group (*p* = 0.005).

The mean symptom duration prior to MRI was 20 months in ALS (SD 14, range 2–49 months) and 70 months in MMN (SD 55, range 13–188 months). Lower motor neuron signs in at least one arm were present in 22/22 patients with ALS and 7/8 patients with MMN, while at least one leg was affected in 22/22 patients with ALS and 3/8 patients with MMN. Individual clinical/neurological characteristics are listed in Appendix B Table A1.

Figure 1 shows representative images and parameter maps of a healthy subject and patients with ALS or MMN. In the upper-arm nerves, patients with MMN showed a decreased FA compared to healthy subjects (0.48 ± 0.08 versus 0.56 ± 0.06; *p* = 0.018; Figure 2). This decrease in FA was due to an increase in RD (1188 ± 233 × 10^−6^ mm^2^/s versus 914 ± 149 × 10^−6^ mm^2^/s; *p* = 0.014), while AD was not statistically different. The nerve CSA of patients with MMN was higher compared to that of both the ALS and healthy controls (9.00 ± 4.74 mm^2^ versus 5.12 ± 0.79 mm^2^ and 4.46 ± 0.78 mm^2^; *p* < 0.001 and *p* < 0.001). Moreover, the T2_app_ in patients with MMN was higher compared to both patients with ALS and to the healthy volunteers (84.6 ± 18.2 ms versus 68.1 ± 6.5 ms and 65.8 ± 6.5 ms; *p* = 0.002 each). The PSD was higher in patients with MMN compared to the control group (324 ± 54 versus 261 ± 39; *p* = 0.029) but not to ALS (*p* = 0.263).

There were no significant differences between ALS patients and healthy subjects in any comparison either at the upper or lower extremity. Neither were there any significant differences in the tibial nerve at the thigh or calf between MMN, ALS, or the control group in any quantitative MR neurography biomarker (*p* > 0.05 each; Figure 3 and Appendix B Figure A1). Values are listed in Appendix B Table A2.

## 4. Discussion

In this study, we assessed the phenotype of MMN and ALS in quantitative MR neurography, using DTI, T2-relaxometry, and conventional nerve CSA. We detected characteristic changes in DTI and T2-relaxometry parameters in the upper-arm nerves of patients with MMN compared to ALS or healthy controls. In contrast, MR neurography biomarkers in ALS patients were not different from healthy subjects.

In MMN patients, we predominantly found changes of DTI parameters in the upper-arm nerves rather than in the tibial nerve, corresponding to the main site of clinical affection. A decrease in FA is generally interpreted as a correlate of structural neuropathy and may be caused by an increase of RD or a decrease of AD or both [28]. In our cohort of MMN patients, this decrease of FA was caused by an increase of RD, which is considered as a marker of demyelination [17] and which was found likewise in both acquired and hereditary demyelinating polyneuropathies, such as chronic inflammatory demyelinating polyneuropathy (CIDP) or Charcot–Marie–Tooth (CMT) type 1 [16,29]. Our findings of a DTI phenotype indicating demyelination are of interest since the pathophysiology of MMN is not yet fully understood. After an initial description as a demyelinating neuropathy, there is also evidence of axonal degeneration and affection of the nodes of Ranvier and the paranodal area [10,30,31,32,33,34]. A decrease of the DTI parameter AD is discussed as a potential marker for axonal neuropathy; however, the evidence for this simple conception is weaker than an increase of RD reflecting demyelination [17]. One reason for this might be that AD may also be modulated by endoneurial edema, and different stages of axonal damage might therefore potentially lead to different phenotypes in DTI imaging [35].

In the MMN patients of our cohort, AD was not decreased compared to both healthy volunteers and patients with ALS, while a previous study using DTI-based tractography of the median and ulnar nerves did find a significant, yet small decrease in AD in MMN patients [18]. This discrepancy may be explained by methodological differences or different stages of disease, although the median duration of symptoms (52 months versus 59 months in our cohort) and IVIG treatment (10/10 versus 6/8 in our cohort) were similar. Another study measuring DTI metrics in cervical roots did not find any significant changes in MMN compared to healthy controls [20], suggesting that peripheral nerves may be more representative regions for MMN affection than plexus fibers. This is also in line with a study of Beecher et al., who reported no hypertrophies in the brachial and lumbosacral plexus of MMN patients [36].

MMN was also characterized by an increase in T2_app_ and PSD, which further hints at demyelination and endoneurial edema. As a quantitative measure of water content, T2-relaxometry detects tissue changes in density and macromolecular composition. Increased T2 relaxation time is a consistent finding in multiple sclerosis as a typically demyelinating disease [37,38] but was also found in amyloidosis, which is accompanied by nerve swelling, while neurodegenerative neuropathies with purely axonal damage showed decreased T2_app_ and PSD [11,12,13,14]. The combination we report also fits well with the consistent finding of fascicular nerve swelling in MMN described by conventional MRI and nerve ultrasound [6,7,8].

In contrast to MMN, ALS is a neurodegenerative disease that is characterized by a loss of axons and myelinated fibers and signs of acute degeneration [39]. Importantly, patients with ALS did not show any differences in imaging parameters of the tibial nerve or the upper-arm nerves in our study. This is in line with two previous investigations of DTI in ALS [18,20] and with the visual assessment based on conventional T2-weighted sequences [6]. Simon et al., on the contrary, reported a significant decrease in FA and AD of the tibial nerve in ALS compared to healthy subjects, which further decreased within 3–6 months of follow-up [19]. Moreover, Lichtenstein et al. reported lower FA values in the sciatic nerves of patients with ALS in comparison with healthy volunteers [21]. However, absolute differences of DTI values were small in both studies, and age differences between patients and healthy controls of 6–8 years might also have contributed to these differences to some extent. Taken together, nerve changes in ALS seem rather subtle or below the level of detection by most currently applied MR neurography sequences.

Limitations of this study are the small number of participants, particularly concerning the group of patients with MMN due to the rarity of the disease. Consequently, it was not possible to further investigate the influence of clinical characteristics, such as disease duration or treatment effects. Similarly, correlations of clinical and electrophysiological findings were beyond the scope of this study but will be examined in subsequent investigations. Due to the heterogenous pattern of affection of upper-arm nerves in patients, we decided on a conservative approach and averaged the imaging measures of the three major arm nerves, which could have decreased the sensitivity of the analysis. Although T2-relaxometry and DTI sequences were shown to be highly reliable and robust [40,41,42], absolute values of quantitative imaging biomarkers may be influenced by a variety of technical parameters and cannot be easily generalized. However, this does not affect the relative differences between the groups that we report, as hardware and sequence parameters were identical in all participants. Care was taken to positioning of the coils and selection of central slices to ensure reproducibility. However, effects due to diffusion gradient inhomogeneity cannot be fully ruled out [43,44,45]. Spectral fat saturation may lead to inhomogeneous fat suppression. Differences in age or weight between groups might affect quantitative MR neurography parameters [24,25,46,47]. In order to control for the younger age and lower body weight of our control group, we used correction formulae based on multiple linear regression, which represents a conservative approach, since this correction rather prevents false-positive results.

## 5. Conclusions

In conclusion, this study indicates a characteristic pattern of quantitative DTI and T2-relaxometry parameters in the upper-arm nerves of patients with MMN, which should be further investigated as biomarkers of early diagnosis and treatment monitoring. Quantitative MR neurography parameters in patients with ALS were not different from healthy controls, implying that nerve pathology in ALS might take place below the detection level of current state-of-the-art quantitative MR neurography.

## Figures and Tables

**Figure 1 diagnostics-13-01237-f001:**
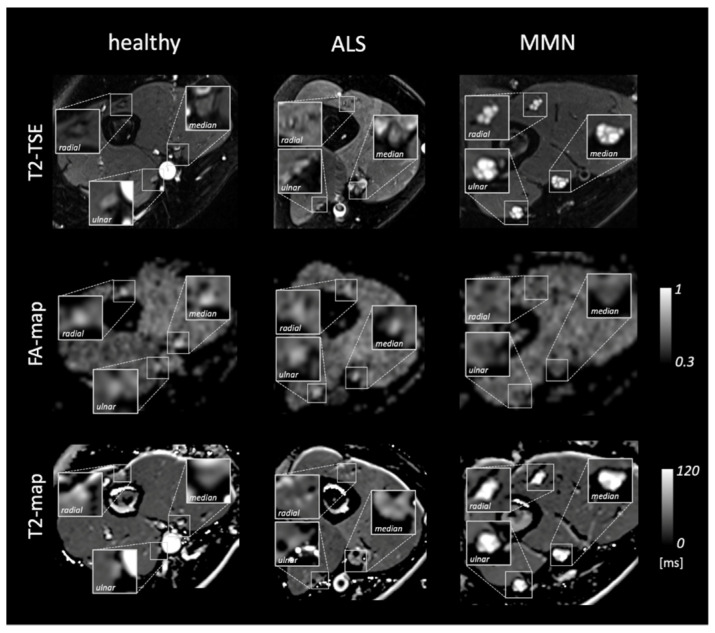
Representative images of a healthy subject and patients with amyotrophic lateral sclerosis (ALS) or multifocal motor neuropathy (MMN). From top to bottom, the rows show representative images of the T2 turbo spin echo (TSE) sequence, the corresponding parameter map of fractional anisotropy (FA-map), and T2-map based on the T2-relaxometry sequence. Insets show a magnification of the median, radial, and ulnar nerves.

**Figure 2 diagnostics-13-01237-f002:**
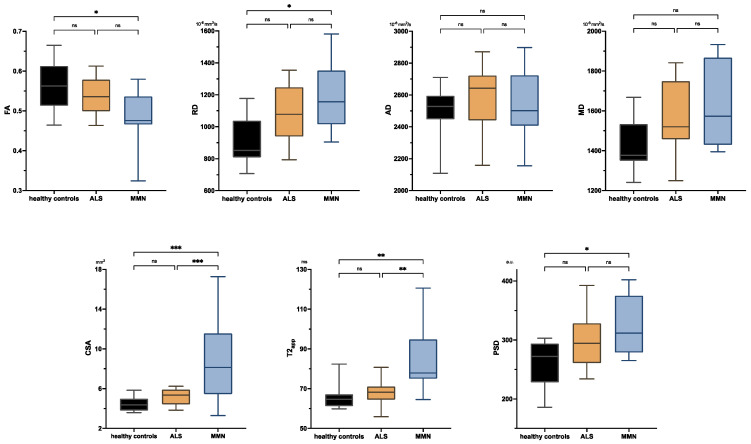
Boxplots of DTI parameters (upper row), cross-sectional area (CSA), and T2-relaxometry metrics (lower row) measured in the upper-arm nerves. Asterisks represent statistical significance at *p* ≤ 0.05 from one-way analysis of variance with Holm–Sidak’s correction. The plots illustrate the 25–75th percentiles (boxes) and minimum-to-maximum values (whiskers). Ns, not significant.

**Figure 3 diagnostics-13-01237-f003:**
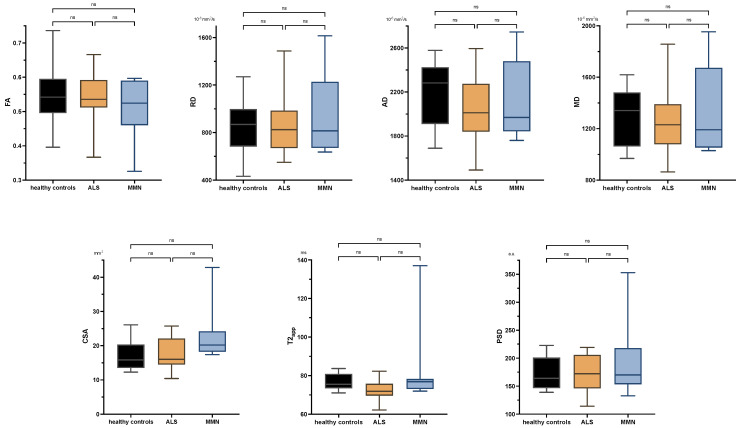
Boxplots of DTI parameters (upper row), cross-sectional area (CSA), and T2-relaxometry metrics (lower row) measured in the tibial nerve at the thigh. One-way analysis of variance with Holm–Sidak’s correction, statistical significance was set at *p* ≤ 0.05. The plots illustrate the 25–75th percentiles (boxes) and minimum-to-maximum values (whiskers). Ns, not significant.

**Table 1 diagnostics-13-01237-t001:** Sequence parameters.

	DTI	T2-Relaxometry	T2w
Repetition time (TR)	3990 ms	2400 ms	8150 ms
Echo time (TE)	72 ms	12 echoes (10, 20, …, 120 ms)	54 ms
Field of view (FOV)	178 × 178 mm^2^	160 × 160 mm^2^	160 × 160 mm^2^
Matrix	102 × 128	338 × 334	512 × 333
Slice thickness	3.0 mm	3.5 mm	3.5 mm
Number of slices	18	11	41
Slice gap	0.3 mm	None	0.35 mm
Number of averages	1	1	2
Parallel imaging	GRAPPA	None	GRAPPA
Acceleration factor	2	n.a.	2
Refocusing flip angle	180°	180°	150°
Bandwidth	930 Hz/px	190 Hz/px	181 Hz/px
Acquisition time per slab	8:36 min	6:43 min	3:56 min
Fat saturation	Spectral fat saturation	Spectral fat saturation	Spectral fat saturation
Number of diffusion directions	20	n.a.	n.a.
b-values	0 and 1000 s/mm^2^	n.a.	n.a.

Abbreviations: DTI, diffusion tensor imaging; T2w, T2-weighted sequence; GRAPPA, generalized autocalibrating partial parallel acquisition.

## Data Availability

The data that support the findings of this study are available upon request from the corresponding author. The data are not publicly available due to restrictions, as they contain information that could compromise the privacy of the research participants.

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
