# Peer review of "Quantitative MR Neurography in Multifocal Motor Neuropathy and Amyotrophic Lateral Sclerosis"

_diagnostics, 2023, doi:10.3390/diagnostics13071237_

Round 1

Reviewer 1 Report (Previous Reviewer 3)

Given previous review history, I accept as is.

Reviewer 2 Report (Previous Reviewer 1)

I stil retain that the robustness of results are limited by the small sample size. 

This manuscript is a resubmission of an earlier submission. The following is a list of the peer review reports and author responses from that submission.

Round 1

Reviewer 1 Report

I read with interest the manuscript by Foesleitner et al on MR neurography in multifocal motor neuropathy and amyotrophic lateral sclerosis. Authors performed a MRN and DTI evaluation of two population of neurological diseases (MMN and ALS) and a group of healthy controls. They found a characteristic imaging pattern in MMN consisting of a decreased FA, increased RD (p=.015), increased CSA (p=.003), increased T2-relaxation time (p<.001), and increased PSD (p=.004) in the upper arm nerves compared to ALS and controls. ALS patients did not differ to controls in any imaging marker.

The study is of potential interest but there are some major issues:

-       In MMN nerve trunks are not equally involved therefore considering the averaged value across different nerves (that may be involved or not) is not appropriate. Moreover, I can’t find the reason to study nerves which are not involved (clinically or neurophysiological).

-       The cohort size is very small, in particular when compared with a similar study (van Rosmalen MHJ et al. Quantitative magnetic resonance imaging of the brachial plexus shows specific changes in nerve architecture in chronic inflammatory demyelinating polyneuropathy, multifocal motor neuropathy and motor neuron disease. Eur J Neurol. 2021) thus preventing the possibility to control for confounding factors (history of diseased, treatment and others).

-       Other studies found different results (van Rosmalen MHJ et al. Quantitative magnetic resonance imaging of the brachial plexus shows specific changes in nerve architecture in chronic inflammatory demyelinating polyneuropathy, multifocal motor neuropathy and motor neuron disease. Eur J Neurol. 2021; Beecher G et al. Plexus MRI helps distinguish the immune-mediated neuropathies MADSAM and MMN. J Neuroimmunol. 2022). In particular they found higher FA in MMN than in ALS, no DTI differences between MMN and controls and normal CSA in MMN. Authors should further explain these differences.

Author Response

Please find our point-by-point response attached.

Reviewer 2 Report

This study aimed to test whether diffusion tensor imaging of the peripheral nerves could be useful in differentiating ALS from MMN. The authors find in MND: lower FA, higher RD, higher T2, higher CSA and higher PSD. While some of these findings were non-significant, they do seem to correspond well to the known underlying pathological changes. The article is well written with no mistakes and the work appears scientifically sound.  The differentiation of ALS and MMN is an important clinical research question. While the study is underpowered, this is partially justified by the rarity of the disorder. I have some minor comments below.

Were the same six slices of the images used for all subjects? If not, couldn’t the cross sectional area change depending on the selected slices, rather than any pathology?

40 subjects were included, but 115 slabs were acquired. Why isn’t there a whole number of slabs per participant?

The analysis could be improved by taking into account the most affected limb.

Given the in-plane resolution and small cross section of the nerves, how badly were the results affected by partial volume effects? Can the authors comment on any methods for partial volume correction and the potential impact?

Author Response

(The authors gave the same response as above.)

Reviewer 3 Report

A well-written manuscript with a lot of potential, but in its current form it is of dubious scientific value (not sufficiently proven, at least for me).

The authors use FA, RD, AD conversion factors depending on age and weight, according to the given formulas, given in other publications and based on groups of 60 people. This is quite a controversial approach for me.

First of all, I would like to see these parameters without conversion factors, i.e. the identical table, suppl.table 2.

Secondly, the DTI metrics strongly depend on the parameters of the experiments, TE, TR, magnetic field, number of b directions, layer thickness, FOV, and finally systematic errors that depend on the hardware (gradient coils, RF coil, MR sequence and its parameters), that is, they are specific to the MR scanner and sequence parameters. This issue is well illustrated by the BSD-DTI technique, described by the Stejskal-Tanner equation for nonuniform magnetic field gradients.

Secondly, how the authors will explain the lack of consideration of the above issues, while focusing on the given conversion factor. After all, there are works that show that FA-type parameters depend, for example, on the way the brain is "used", e.g. by professional musicians.

Another question, how are PSD images normalized, what is the norm? While in DTI, the tensor after diagonalization is related to the microstructure, here what was the model norm for PSD?

Author Response

(The authors gave the same response as above.)

Round 2

Reviewer 1 Report

Authors ameliorated the manuscript. Nonetheless, the small size of the cohorts prevents the solidity of the results.

Reviewer 3 Report

I mostly accept the explanation, it could be improved, but that would be "an academic discussion".

There is one thing I cannot agree with, at least not entirely.

"However, this does not affect the relative differences between groups we report, as hardware and sequence parameters were identical in all participants."

This is not entirely true, in part it is.

The systematic error (described by the BSD-DTI problem, and the Generalized ST Equation for nonuniform magnetic field gradients) depends on the location in space, so we have to watch, in addition to the sequence parameters, also the identical location in the magnet, in the laboratory frame. Otherwise, systematic errors may add differently and not average in the same way. And the difference between the groups can be highlighted by systematic errors that did not average.

If the segmented ROIs for patients and controls hit the same areas in the magnet space (laboratory frame) then your assumption may be true that there will be little impact on differential data (numerically); please explain here.

Another issue is the impact on the result in the context of statistically significant differences (SSD). Systematic error, like white noise, increases the bias of the DTI metrics, so an SSD could look very different...

So, I suggest considering the above 2 issues (probably best in BSD context...), they seem to be important.

These are the last important issues that require clarification. The manuscript is now much clearer.